# Providing Antibacterial Activity to Poly(2-Hydroxy Ethyl Methacrylate) by Copolymerization with a Methacrylic Thiazolium Derivative

**DOI:** 10.3390/ijms19124120

**Published:** 2018-12-19

**Authors:** Alexandra Muñoz-Bonilla, Daniel López, Marta Fernández-García

**Affiliations:** Instituto de Ciencia y Tecnología de Polímeros (ICTP-CSIC), C/ Juan de la Cierva 3, 28006 Madrid, Spain; daniel@ictp.csic.es

**Keywords:** antimicrobial polymers, quaternary ammonium, 2-hydroxyethyl methacrylate, thermal stability

## Abstract

Antimicrobial polymers and coatings are potent types of materials for fighting microbial infections, and as such, they have attracted increased attention in many fields. Here, a series of antimicrobial copolymers were prepared by radical copolymerization of 2-hydroxyethyl methacrylate (HEMA), which is widely employed in the manufacturing of biomedical devices, and the monomer 2-(4-methylthiazol-5-yl)ethyl methacrylate (MTA), which bears thiazole side groups susceptible to quaternization, to provide a positive charge. The copolymers were further quantitatively quaternized with either methyl or butyl iodide, as demonstrated by nuclear magnetic resonance (NMR) and attenuated total reflection Fourier-transform infrared spectroscopy (ATR-FTIR). Then, the polycations were characterized by zeta potential measurements to evaluate their effective charge and by differential scanning calorimetry (DSC) and thermogravimetric analysis (TGA) to evaluate their thermal properties. The ζ-potential study revealed that the quaternized copolymers with intermediate compositions present higher charges than the corresponding homopolymers. The cationic copolymers showed greater glass transition temperatures than poly(2-hydroxyethyl methacrylate) (PHEMA), with values higher than 100 °C, in particular those quaternized with methyl iodide. The TGA studies showed that the thermal stability of polycations varies with the composition, improving as the content of HEMA in the copolymer increases. Microbial assays targeting Gram-positive and Gram-negative bacteria confirmed that the incorporation of a low number of cationic units into PHEMA provides antimicrobial character with a minimum inhibitory concentration (MIC) of 128 µg mL^−1^. Remarkably, copolymers with MTA molar fractions higher than 0.50 exhibited MIC values as low as 8 µg mL^−1^.

## 1. Introduction

In the last few years, antimicrobial polymers have attracted substantial scientific and industrial attention because of their unique properties and applications in the design and production of many materials, including medical devices, textiles, packaging, and purification systems [1,2,3]. Of special concern is bacterial contamination on the surfaces of medical devices, such as catheters or implants, which are responsible for many hospital-acquired infections (HAI), also known as nosocomial infections. The most common and serious HAIs are catheter-associated urinary tract infections, central line-associated bloodstream infections, ventilator-associated pneumonia, and surgical site infections, among others [4,5]. Coatings able to eliminate bacterial contamination on these material surfaces and, thus, prevent such infections have emerged as very efficient prophylactic strategies. These coatings can either repel microbes, avoiding microbial attachment, or kill microorganisms upon contact or in the surrounding by biocidal release [6]. In particular, self-disinfecting coatings with killing or bactericidal activity have demonstrated high efficiency and are typically obtained by incorporating antimicrobial agents, including antibiotics [7,8], antimicrobial peptides [9,10], silver and copper compounds [11,12], zinc oxide [13], titanium dioxide particles [14,15], etc., onto the surface. Alternatively, the use of antimicrobial polymers as contact-active coatings with inherent biocidal activity has gained importance, because antimicrobial polymers offer some advantages, such as chemical stability, high and long-term activity, low toxicity, and reduced potential to generate resistance [16,17]. Most of these antimicrobial polymers are polycations, in particular polymers with quaternary ammonium groups [18,19], which are able to interact with the negatively charged bacterial wall, disrupting the integrity of the membrane and leading to the death of the bacteria. Recently, we developed a series of polycations based on polymethacrylates bearing pendant 1,3-thiazolium groups, which have demonstrated high activity against a broad spectrum of bacteria [20,21,22]. In addition, antimicrobial polymeric coatings have been prepared from these polycations by blending with hydrophobic polymers typically used in medical devices, such as polyacrylonitrile or polystyrene [23,24,25]. 

Concerning hydrophilic polymer materials, poly(2-hydroxyethyl methacrylate) (PHEMA) is one of the most widely used in the manufacture of medical devices, such as contact and intraocular lenses, and medical device coatings [26,27], because it exhibits blood and cell biocompatibility, low cytotoxicity, and thrombogenicity [28,29]. In addition, a variety of antimicrobial agents has been included into hydrogels and materials based on PHEMA to provide biocidal character as an additional property [26,30,31,32]. Herein, we prepared copolymers systems composed of 2-hydroxyethyl methacrylate (HEMA) and a methacrylic monomer bearing thiazolium moieties (MTARI) with the purpose of incorporating antimicrobial properties into PHEMA. Several copolymer compositions were prepared and evaluated for an adequate balance of structural, thermal, and antimicrobial properties.

## 2. Results and Discussion

### 2.1. Synthesis of Cationic Polyelectrolytes: P(MTARI-co-HEMA) Copolymers

First, the synthesis of P(MTA-*co*-HEMA) copolymers was performed by free radical polymerization of HEMA and 2-(4-methylthiazol-5-yl)ethyl methacrylate (MTA) comonomers, using different feed molar ratios, i.e., feed molar fraction of MTA, f_MTA_ = 0.0, 0.2, 0.4, 0.6, 0.8, and 1.0 (Scheme 1). 

The copolymerizations almost reached full conversion after 24 h for all the initial f_MTA_, which was confirmed gravimetrically and by ^1^H-NMR (the double bonds completely disappeared from the bulk of the reactions). Similarly, the molar fractions of MTA in the obtained copolymers (F_MTA_) were determined by ^1^H-NMR, and as expected, these values were found to be very close to the f_MTA_ values (Table 1) as the conversion was nearly complete (f_MTA_ ≅ F_MTA_). Table 1 summarizes the average molecular weight (M_n_) and the polydispersity indexes (PDI) of the P(MTA-*co*-HEMA) copolymers determined by gel permeation chromatography (GPC). The molecular weights ranged from 35 to 87 kDa, while the PDI values were around 1.9–2.4, similar to those typically obtained in radical polymerization. 

Subsequently, the corresponding polycations with different charge balances were prepared by *N*-alkylation of the P(MTA-*co*-HEMA) copolymers with either butyl or methyl iodide, as shown in Scheme 2. 

To ensure the complete quaternization of all the thiazole groups present in the copolymers, the reaction was carried out with an excess of the alkylating agent at 70 °C. After one week, complete quaternization was reached for all the cases as revealed by ^1^H-NMR spectra. As an example, Figure 1 depicts the spectra of the quaternized copolymers with butyl iodide, P(MTABuI-*co*-HEMA), and their corresponding homopolymers, PHEMA and PMTABuI. The MTA homopolymer (PMTA) spectrum is also depicted for comparative purposes to visualize the shift alteration produced by the protonation. It can be clearly seen that the signals corresponding to the aromatic protons of 1,3-thiazole, –N=CH–S, at ~8.8 ppm shifted to ~10.1–10.2 ppm after the *N*-alkylation to obtain 1,3-thiazolium group, –N^+^=CH–S, which confirmed that all the modifications were achieved quantitatively [33,34]. In addition, new signals appeared at ~4.4 ppm due to the alkylating agent (i.e., –N^+^–CH_2_– in the case of butyl iodide). The intensity of this signal increased as the content of MTA increased in the copolymer, that is, with increasing values of F_MTA._ This event was concomitant with a decrease in the intensity of the signals attributed to the HEMA units, such as the peak at 4.8 ppm that corresponded to the proton of the hydroxyl group. The copolymers were also characterized by ATR-FTIR spectroscopy. As an example, Figure 2 shows the spectra of the unquaternized and quaternized copolymers with both alkylating agents with an active comonomer composition of 0.8, *viz*. P(MTA_0.8_-*co*-HEMA_0.2_), P(MTAMeI_0.8_-*co*-HEMA_0.2_), and P(MTABuI_0.8_-*co*-HEMA_0.2_), respectively. The carbonyl stretching vibration (C=O) at around 1720 cm^−1^, characteristic of methacrylic monomers, the O–H stretching region around 3700–3100 cm^−1^, and the C–O band at ca. 1250 cm^−1^, typical of HEMA polymers, can be observed clearly in the spectra. The band corresponding to the C=N– stretching vibration of MTA appeared at 1550 cm^−1^. This band vanished when the copolymers were modified with the alkyl iodine agents, and a new band emerged at ca. 1590 cm^−1^, characteristic of the C=N^+^– stretching vibration. 

### 2.2. Characterization of the Synthetized Copolymers: P(MTA-co-HEMA), and P(MTARI-co-HEMA)

Once the copolymer precursors, P(MTA-*co*-HEMA), and the polycations P(MTARI-*co*-HEMA) were successfully prepared, they were characterized to estimate their antimicrobial potential. It is well known that such activity is dependent on different parameters, such as the nature of the charge; the hydrophobic groups; the balance of cationic to hydrophobic moieties; the polymer composition; and the molecular weight [1,35,36]. Then, the ζ-potential of the polycations was determined, and the obtained values are represented in Figure 3.

Interestingly, the compositions with higher contents of MTA presented greater zeta potential values than their corresponding quaternized homopolymers. They presented values near to +60 or higher, which indicated the good stability of the aggregation in comparison with the homopolymers, especially with respect to the PMTAMeI, which has moderate stability with a ζ-potential of +40. In fact, the PMTABuI homopolymer was more stable and positively charged than the PMTAMeI: therefore, higher antimicrobial activity was to be expected. 

Subsequently, the thermal properties of the obtained copolymers were analyzed given that they were of great importance for the applicability of these antimicrobial polymers. The copolymers were first analyzed by differential scanning calorimetry (DSC), and the glass transition temperatures, T_g_, are given in Table 2. Figure 4a displays the DSC curves of the unquaternized copolymers, while Figure 4b represents the T_g_ variation of all the series as a function of the copolymer composition, F_MTA_.

From these data, it was observed that the T_g_ values of the copolymers shifted to lower temperatures as the content of MTA, F_MTA_, increased up to 49 °C for the PMTA homopolymer. In contrast, the polycations obtained after quaternization followed the contrary trend; their T_g_ increased with the amount of the MTARI cationic units for both series of copolymers—those quaternized with methyl iodide (P(MTAMeI-*co*-HEMA)) and those quaternized with butyl iodide (P(MTABuI-*co*-HEMA)). Also, both series of cationic copolymers exhibited greater T_g_ than the PHEMA homopolymer. It was noticeable that the P(MTAMeI-*co*-HEMA) copolymers achieved higher T_g_ values in comparison with the P(MTABuI-*co*-HEMA) series due to the effect of the length of the alkylating agent. The incorporation of a long and flexible alkyl chain, such as a butyl group, improved the mobility of the copolymers and reduced their T_g_.

Then, the thermal stability of the different series was analyzed by TGA under an inert atmosphere. Figure 5 displays the TGA curves of the unquaternized and quaternized copolymer, and the thermal degradation parameters are collected in Table 2. The degradation of PHEMA took place in one single stage, considering that a previous step of water elimination occurred because of the hygroscopic character of the polymer (ca. 2–3%). The literature has explained that the resulting product of this breakdown is mainly the HEMA monomer [37,38,39]. PMTA presented two main stages, and contrary to HEMA, in which depolymerization was the main process, the degradation seemed to be by random chain scission as with poly(methyl methacrylate). In the case of the unmodified copolymers, the behavior was dependent on the HEMA/MTA content. Those copolymers with HEMA predominance presented hygroscopic tendencies and intermediate behaviors between both homopolymer parents. The first main stage, at temperatures higher than 300 °C, is shifted to higher temperatures as the MTA increased, while in the second step, the temperatures decreased. Nevertheless, the stability was improved by the presence of HEMA in the copolymer. 

This behavior was more evident in both the quaternized copolymer series. Figure 5b,c show that the incorporation of the HEMA units into the copolymers expanded the thermal stability, which could extend the applicability of these antimicrobial materials. In the case of methyl iodide incorporation, the degradation occurred in three stages instead of the two appearing in the case of butyl iodide. Therefore, the quaternization with a longer alkyl agent stabilized the macromolecular structure of the cationic copolymers. In contrast, the polycations obtained after the quaternization followed the contrary trend; their T_g_ increased with the amount of the MTARI cationic units for both series of copolymers—those quaternized with methyl iodide (P(MTAMeI-*co*-HEMA)) and those quaternized with butyl iodide (P(MTABuI-*co*-HEMA)). Also, both series of cationic copolymers exhibited greater T_g_ than the PHEMA homopolymer, whose value was almost 100 °C due to the strong inter- and intramolecular interactions [37,40]. It is noticeable that the P(MTAMeI-*co*-HEMA) copolymers achieved higher T_g_ values in comparison with the copolymers quaternized with butyl iodide because of the effect of the length of the alkylating agent. The incorporation of a long and flexible alkyl chain, such as a butyl group, improved the mobility of the copolymers and reduced their T_g_.

### 2.3. Antibacterial Activity Studies

The antimicrobial activity of the prepared polycations, the P(MTAMeI-*co*-HEMA) and P(MTABuI-*co*-HEMA) copolymers, was evaluated against the model bacterial strains, Gram-positive *Staphylococcus aureus* and Gram-negative *Pseudomonas aeruginosa*. These polymers were also tested against fungi *Candida parapsilosis*, but they were not effective in the opposite behavior when MTA was copolymerized with acrylonitrile, a hydrophobic monomer [21]. Concretely, the microbroth dilution reference method [41,42,43] was used, obtaining the minimum inhibitory concentration (MIC) values collected in Table 3. As expected, the homopolymers, PMTAMeI and PMTABuI, exhibited significant antimicrobial activity with very low MIC values, as previously reported [20]. The homopolymer quaternized with butyl iodide showed improved activity against the Gram-negative bacteria in comparison with the methylated polymer, because it augments the hydrophobic balance of the copolymers [35]. In effect, several studies have demonstrated that the incorporation of certain contents of hydrophobic moieties, reaching an adequate hydrophobic/hydrophilic balance, improves the antimicrobial activity of the polymer, because the process facilitates the pass through the hydrophobic cytoplasmic membrane [44,45].

On the other hand, when biocompatible HEMA units are incorporated into the copolymer, the activity tends to diminish. This is due to the decrease in the positive charge density of the corresponding polycations as a result of the incorporation of a non-active monomer. Nevertheless, the copolymers mainly based on HEMA, with MTARI contents as low as F_MTA_ = 0.18, still exhibited significant activity—MIC values of 128 µg mL^−1^. Again, the copolymers containing butyl groups showed better activities than the copolymers quaternized with methyl iodide. Remarkably, the copolymers quaternized with butyl iodide maintained their excellent activity up to a relatively high content of HEMA, with MIC = 8 µg mL^−1^ for the F_MTA_ value of 0.56. In this case, the copolymers might have adopted in-solution conformations in which their positive charges were highly accessible to bacterial membrane. 

Therefore, the obtained copolymers were demonstrated to be promising antimicrobial materials, in which the incorporation of even a low number of cationic units into PHEMA provided significant antibacterial activity and maintained good thermal stability. While higher amounts of the cationic monomer, up to ~50%, maintained the excellent antimicrobial activity, reaching MIC values similar to that found in the homopolymers PMTARI, and improved their thermal stability, the monomer could also enhance their biocompatibility, because PHEMA and PMTARI are not toxics [20,22,28,29].

## 3. Materials and Methods

### 3.1. Materials

The monomer 2-(4-methylthiazol-5-yl)ethyl methacrylate (MTA) was synthesized as previously reported [20]. The monomer 2-hydroxyethyl methacrylate (HEMA, 99%; Aldrich, Steinheim, Germany) was distilled prior to use. 2,2′-Azobisisobutyronitrile (AIBN, 98%; Acros, Buch, Switzerland) was recrystallized twice from methanol (MeOH, 99.9%; Aldrich) prior to use. Anhydrous dimethyl sulfoxide (DMSO, 99.8%) and *N*,*N*-dimethylformamide (DMF, 99.8%) were purchased from Alfa-Aesar (Karlsruhe, Germany) and were used as received. 1-Iodobutane (BuI, 99%, Aldrich), iodomethane (MeI, 99%; Aldrich), and hexane (96%; Scharlau, Sentmenat, Spain) were used as received. 

For the microbiological assays, sodium chloride (NaCl, 0.9%, BioXtra, Steinheim, Germany, suitable for cell cultures) and phosphate buffered saline (PBS, pH 7.4) were obtained from Aldrich. BBL^TM^ Mueller Hinton broth used as microbial growth media was purchased from Becton, Dickinson and Company (Madrid, Spain). Sheep blood (5%) and Columbia Agar plates were acquired from BioMérieux (Madrid, Spain). American Type Culture Collection (ATCC) Gram-positive *Staphylococcus aureus* (*S. aureus*, ATCC 29213) and Gram-negative *Pseudomonas aeruginosa* (*P. aeruginosa*, ATCC 27853) bacteria were obtained from Oxoid^TM^ (Wesel, Germany).

### 3.2. Synthesis of P(MTA-co-HEMA) Copolymers

P(MTA-*co*-HEMA)s copolymers with different chemical compositions were synthesized via free radical polymerization of HEMA and MTA comonomers, as shown in Scheme 1. Briefly, both monomers, MTA and HEMA (1 M total concentration), and the initiator, AIBN (5 × 10^−2^ M), were added into a Schlenk tube and dissolved in anhydrous DMSO. The mixture was deoxygenated by purging with argon over 15 min. Then, the reaction was stirred at 60 °C for 20 h under an argon atmosphere. After that, the mixture was cooled down, and the polymers were isolated by precipitation into distilled water, filtered, and washed several times with water. The solid was dried under a vacuum until a constant weight was reached. 

P(MTA-*co*-HEMA): ^1^H-NMR (300 MHz, DMSO-d6): δ = 8.80–8.85 (br, 1H; =CH thiazole, MTA), 4.80 (br, 1H; OH, HEMA), 4.02 (br, 2H; OCH**_2_**, MTA), 3.90 (br, 2H; –C**H_2_**–OH, HEMA), 3.59 (br, 2H; –CH**_2_**–CO–, HEMA), 3.05 (br, 2H; CH**_2_**, MTA), 2.30 (br, 3H; CH**_3_** thiazole, MTA), 1.92–1.24 (br, 2H; CH**_2_**, MTA), 0.76–0.57 (br, 3H; CH**_3_**, MTA). 

### 3.3. Quaternization of Copolymers: Synthesis of Cationic Polyelectrolytes, P(MTARI-co-HEMA)

The P(MTA-*co*-HEMA) copolymers were modified by *N*-alkylation of the thiazole groups of the MTA units with 1-iodobutane or iodomethane, as shown in Scheme 2. The copolymers were added into a sealed tube containing a magnetic stirring bar and dissolved in anhydrous DMF (0.1 mmol L^−1^). Then, a large excess of alkyl iodide, methyl iodide, or butyl iodide was added (ratio copolymer/alkyl iodide ≈ 1:5). The mixture was purged with argon and heated at 70 °C while being stirred for one week to ensure complete quaternization. Then, the solution was poured into hexane, and the copolymers were obtained as brown oils. The quaternized copolymers were further purified by dialysis against the distilled water to remove the residual products and were freeze dried. The methylated and butylated copolymers were labeled as P(MTAMeI-*co*-HEMA) and P(MTABuI-*co*-HEMA), respectively. The degree of quaternization was determined by ^1^H-NMR spectroscopy [22].

P(MTAMeI-*co*-HEMA): ^1^H-NMR (300 MHz, DMSO-d6): δ = 10.16–10.06 (br, 1H; =CH thiazolium, MTAMeI), 4.80 (br, 1H; OH, HEMA), 4.20–3.99 (br, 5H; ^+^NCH**_3_** and OCH**_2_**, MTAMeI), 3.90 (br, 2H; –C**H_2_**–OH, HEMA), 3.59 (br, 2H; –CH**_2_**–CO–, HEMA), 3.40 (br, 2H; CH_2_, MTAMeI), ~2.57 (br, 3H; CH**_3_** thiazolium, MTAMeI), 2.14–1.47 (br, 2H; –CH**_2_**–, MTAMeI), 1.80 (br, 2H, –CH**_2_**–, HEMA), 1.10–0.60 (br, 3H; CH**_3_**, HEMA) 1.07–0.41 (br, 3H; CH**_3_**, MTAMeI). 

P(MTABuI-*co*-HEMA): ^1^H-NMR (300 MHz, DMSO-d_6_): δ = 10.27–10.10 (br, 1H; =CH thiazolium, MTABuI), 4.80 (br, 1H; OH, HEMA), 4.56–4.45 (br, 2H; ^+^NCH**_2_**, MTABuI), 4.09–4.02 (br, 2H; OCH**_2_**, MTABuI), 3.90 (br, 2H; –C**H_2_**–OH, HEMA), 3.59 (br, 2H; –CH**_2_**–CO–, HEMA), 3.32 (br, 2H; CH**_2_**, MTABuI), ~2.57 (br, 3H; CH**_3_** thiazolium, MTABuI), 1.93–1.65 (br, 2H; CH**_2_**, MTABuI), 1.80 (br, 2H, –CH**_2_**–, HEMA), 1.10–0.60 (br, 3H; CH**_3_**, HEMA),1.48–1.23 (br, 4H; 2CH**_2_**, MTABuI), 0.98–0.34 (br, 6H; 2CH**_3_**, MTABuI).

### 3.4. Characterization Methods

The ^1^H nuclear magnetic resonance (NMR) spectra were recorded on a Varian System-500 at room temperature using deuterated chloroform (CDCl_3_) and DMSO-d6 purchased from Sigma-Aldrich as solvents. Fourier-transform infrared (FTIR) spectra were recorded on a Perkin Elmer Spectrum Two instrument with a high-performance, room temperature LiTaO_3_ (lithium tantalate) detector and a universal attenuated total reflectance (ATR) instrument with a diamond/ZnSe crystal. The absorptions were given in wavenumber (cm^−1^), and the spectrum was performed in scan range from 4000 to 450 cm^−1^ with a 0.5 cm^−1^ resolution and 16 scans. The molecular weights and polydispersity indexes of the synthetized copolymers were determined by gel permeation chromatography (GPC) on a Waters Division Millipore system and a Waters 2414 refractive index detector with a 1 mL/min^-1^ flow rate of DMF (GPC-grade, stabilized with 0.1 M LiBr, Scharlau) as eluent at 50 °C. The calibration was performed with poly(methyl methacrylate) standards (Polymer Laboratories LTD). The zeta potential measurements were conducted using the Zetasizer Nano series ZS (Malvern Instruments Ltd, Malvern, UK). The zeta potential of the polymers in deionized water was an average of 10 measurements. Differential scanning calorimetry (DSC) measurements were conducted on a TA Q2000 instrument under dry nitrogen (50 cm^3^ min^−1^). The samples were equilibrated at −70 °C and heated to 120 °C at 10 °C/min. Then, they were cooled to −70 °C and again heated to 120 °C at similar scanning rates. The thermogravimetric analysis (TGA) of the copolymers was performed on a TA Instrument (TGA Q500, TA Instruments, New Castle, Delaware, US) at a heating rate of 10 °C/min^−1^ from 40 to 800 °C under a nitrogen atmosphere. The instrument was calibrated both for temperature and weight by standard methods.

### 3.5. Microbial Growth Inhibition Assays

The antimicrobial activity of the quaternized copolymers was tested against the ATCC microbial strains according to the Clinical Laboratory Standards Institute (CLSI) microbroth dilution reference methods [42,43,46]. The microorganisms were incubated on 5% sheep blood and Columbia Agar plates (BioMérieux) for 24 h at 37 °C in a Jouan IQ050 incubator (Winchester, VA, USA). Then, the microorganism concentration was adjusted with a saline solution to a turbidity equivalent to ca. 0.5 of the McFarland turbidity standard, which corresponds to about 10^8^ colony-forming units (CFU) mL^−1^. The optical density of the microorganism suspensions was measured in a DensiCHEK^TM^ Plus (VITEK, BioMérieux). These suspensions were further diluted with Mueller–Hinton broth to obtain 2 × 10^6^ CFU mL^−1^. The copolymers were dissolved in a mixture of sterile water and a minimum amount of DMSO (up to 6% *v*/*v* as a higher DMSO content was demonstrated to be toxic for these bacterial strains [20,47]) to obtain solutions of 256 μg mL^−1^ for each copolymer. Then, the broth microdilution method was carried out as follows: 100 μL of each copolymer solution was placed in the first column of a 96-well round-bottom microplate. Subsequently, 50 μL of broth was added into the rest of the wells (except in the first column). In the first column, 50 μL of the copolymer solution was diluted by 2-fold serial dilutions in the rest of the wells, and finally, all the wells of the microdilution plates were inoculated with 50 μL of each test microorganism sample to yield a total volume of 100 μL, bacterial concentrations of 5 × 10^5^ CFU mL^−1^, and copolymer concentrations of 128, 64, 32, 16, 8, 4, 2, 1, 0.5, 0.25, and 0.125 μg mL^−1^. Positive and negative controls were also performed. The plates were incubated at 37 °C for 24 h, and the MIC was visually determined to be the lowest concentration of the antimicrobial copolymer in which no bacterial growth was observed. The tests were performed in triplicate.

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
