# Peer review of "Providing Antibacterial Activity to Poly(2-Hydroxy Ethyl Methacrylate) by Copolymerization with a Methacrylic Thiazolium Derivative"

_ijms, 2018, doi:10.3390/ijms19124120_

Reviewer 1 Report

The authors present another (cf. refs. nos. 14–18) antimicrobial polymers preparation and basic characterization concerning their physico-chemical and antimicrobial properties. Such antimicrobial application is interesting and could be applied in industry, especially in biomedical applications. The manuscript is well written but the data presentation must be improved and inaccuracies must be corrected.

 The following paragraphs briefly summarize recommended improvements, changes, and errors of the manuscript's parts:

 Abstract

– The abbreviation PHEMA is not defined in the abstract (independently of the list of abbreviations).

 Results and Discussions

– lines from 87 to 107 – The band assignment of NMR and FTIR shifts/bands are not properly referenced. Even if the band assignments are well known, some standard references (similarly to your previous papers) must be given.

 Materials and Methods

– 3.4. Characterization – The ATR-FTIR experiment is explained in insufficient detail. ATR optical material, FTIR resolution, type of the detector and no. of scans must be added at least.

– lines 299–300 – the given DMSO limit of toxicity is not properly referenced. In the given references nos. 14, 39 are no experiments concerning blank experiments using DMSO. In the paper, e.g.,[T Wadhwani, K Desai, D Patel, D Lawani, P Bahaley, P Joshi, V Kothari. Effect of various solvents on bacterial growth in context of determining MIC of various antimicrobials. The Internet Journal of Microbiology. 2008 Volume 7 Number 1] is shown that at 6 %, DMSO is relatively toxic, it is not so harmless as is claimed in the paper, at least.

 Abbreviations

– PHEMA, used in the introduction, and PMTA in variants, used in the results and discussion section, are missing in the list of abbreviations.

 References

– The references must be arranged according to the instructions for authors, i.e., abbreviated journal names must be used, see refs. nos. 2, 12, 31. Full points are missing in the abbreviated journal titles in refs. nos. 7, 11, 17, 20, 22, 24, 25, 36. The pages are missing in refs. nos. 3, 18, 19. Uppercase letter of “NO” is appropriate in ref. no. 24.

 Figures

– Figure 1 could be improved by adding ppm values of the prominent NMR bands.

– Figure 2 could be also improved by introducing of wavenumbers of the FTIR bands cited in the text.

Author Response

The authors present another (cf. refs. nos. 14–18) antimicrobial polymers preparation and basic characterization concerning their physico-chemical and antimicrobial properties. Such antimicrobial application is interesting and could be applied in industry, especially in biomedical applications. The manuscript is well written but the data presentation must be improved and inaccuracies must be corrected.

 The following paragraphs briefly summarize recommended improvements, changes, and errors of the manuscript's parts:

Abstract

– The abbreviation PHEMA is not defined in the abstract (independently of the list of abbreviations).

First al all, we would like to thank the reviewers for their carefully reading and their suggestions.

PHEMA is now defined in the abstract.

 Results and Discussions

– lines from 87 to 107 – The band assignment of NMR and FTIR shifts/bands are not properly referenced. Even if the band assignments are well known, some standard references (similarly to your previous papers) must be given.

Proper references have been included.

 Materials and Methods

– 3.4. Characterization – The ATR-FTIR experiment is explained in insufficient detail. ATR optical material, FTIR resolution, type of the detector and no. of scans must be added at least.

The ATR-FTIR has been explained adequately.

– lines 299–300 – the given DMSO limit of toxicity is not properly referenced. In the given references nos. 14, 39 are no experiments concerning blank experiments using DMSO. In the paper, e.g.,[T Wadhwani, K Desai, D Patel, D Lawani, P Bahaley, P Joshi, V Kothari. Effect of various solvents on bacterial growth in context of determining MIC of various antimicrobials. The Internet Journal of Microbiology. 2008 Volume 7 Number 1] is shown that at 6 %, DMSO is relatively toxic, it is not so harmless as is claimed in the paper, at least.

The references introduced in the text were 14 and 41 (new references 20 and 47, in the revised version) and these works have been done by our group and we did the blank every time and it is described in the articles. In the article that the reviewer have suggested, “The Internet Journal of Microbiology, 2008 Volume 7 Number 1”, DMSO exerts toxicity up to 4%, the worse values that we present have as maximum content of 3% (we are doing dilutions), and in the case of MIC = 8 μg mL−1 the concentration should be 0.37% that is not toxic at all. In any case, we always perform positive and negative controls to ensure the data.

 Abbreviations

– PHEMA, used in the introduction, and PMTA in variants, used in the results and discussion section, are missing in the list of abbreviations.

We apologize for the mistake. The list has been updated.

 References

– The references must be arranged according to the instructions for authors, i.e., abbreviated journal names must be used, see refs. nos. 2, 12, 31. Full points are missing in the abbreviated journal titles in refs. nos. 7, 11, 17, 20, 22, 24, 25, 36. The pages are missing in refs. nos. 3, 18, 19. Uppercase letter of “NO” is appropriate in ref. no. 24.

These have been corrected.

 Figures

– Figure 1 could be improved by adding ppm values of the prominent NMR bands.

– Figure 2 could be also improved by introducing of wavenumbers of the FTIR bands cited in the text.

The figures have been changed.

Reviewer 2 Report

Dear Authors 

This manuscript idea is quite novel and very attractive topic nowadays.  Lots of paper reporting currently for the modification of biomaterials with antimicrobial agents. This all due to high antibiotic resistance in all around the world. 

Authors work is highly appreciated but needs some expert amendments and editing require.

Line 45: In this line, authors can extract information from these papers and cite them. It will increase the recent experiments and papers on this topic. 

a)  Kelly, Micah, et al. "Peptide aptamers: novel coatings for orthopaedic implants." Materials Science and Engineering: C54 (2015): 84-93.

b) Khurshid, Z., Naseem, M., Sheikh, Z., Najeeb, S., Shahab, S., & Zafar, M. S. (2016). Oral antimicrobial peptides: Types and role in the oral cavity. Saudi Pharmaceutical Journal24(5), 515-524.

In result and discussion heading try to cite the following papers;

a) Rodriguez, G. M., Bowen, J., Grossin, D., Ben-Nissan, B., & Stamboulis, A. (2017). Functionalisation of Ti6Al4V and hydroxyapatite surfaces with combined peptides based on KKLPDA and EEEEEEEE peptides. Colloids and Surfaces B: Biointerfaces160, 154-160.

b) Khurshid, Z., Najeeb, S., Mali, M., Moin, S. F., Raza, S. Q., Zohaib, S., ... & Zafar, M. S. (2017). Histatin peptides: Pharmacological functions and their applications in dentistry. Saudi Pharmaceutical Journal25(1), 25-31.

Line 261-271: These lines need serious attention for the better reading for the readers.  

Heading 3.4 and 3.5: is not well written. Look like authors don't take it a serious part of the papers. Try to redesign it according to scientific writing art. 

This paper also needs some English editing company for the betterment. 

Author Response

Dear Authors

This manuscript idea is quite novel and very attractive topic nowadays.  Lots of paper reporting currently for the modification of biomaterials with antimicrobial agents. This all due to high antibiotic resistance in all around the world.

Authors work is highly appreciated but needs some expert amendments and editing require.

Line 45: In this line, authors can extract information from these papers and cite them. It will increase the recent experiments and papers on this topic.

a)  Kelly, Micah, et al. "Peptide aptamers: novel coatings for orthopaedic implants." Materials Science and Engineering: C54 (2015): 84-93.

b) Khurshid, Z., Naseem, M., Sheikh, Z., Najeeb, S., Shahab, S., & Zafar, M. S. (2016). Oral antimicrobial peptides: Types and role in the oral cavity. Saudi Pharmaceutical Journal, 24(5), 515-524.

In result and discussion heading try to cite the following papers;

a) Rodriguez, G. M., Bowen, J., Grossin, D., Ben-Nissan, B., & Stamboulis, A. (2017). Functionalisation of Ti6Al4V and hydroxyapatite surfaces with combined peptides based on KKLPDA and EEEEEEEE peptides. Colloids and Surfaces B: Biointerfaces, 160, 154-160.

b) Khurshid, Z., Najeeb, S., Mali, M., Moin, S. F., Raza, S. Q., Zohaib, S., ... & Zafar, M. S. (2017). Histatin peptides: Pharmacological functions and their applications in dentistry. Saudi Pharmaceutical Journal, 25(1), 25-31.

First al all, we would like to thank the reviewers for their carefully reading and their suggestions.

Indeed these works are very important and certainly clarifying. We have incorporated two references in the introduction since we mentioned general coatings. However, in the results and discussion is difficult to compare the activity of antimicrobial peptides with polymers. Therefore, we have considered not discussing about antimicrobial peptides since it is not the subject of our work.

Line 261-271: These lines need serious attention for the better reading for the readers. 

The NMR description is done according to standards on elucidation of organic compounds.

Heading 3.4 and 3.5: is not well written. Look like authors don't take it a serious part of the papers. Try to redesign it according to scientific writing art.

This paper also needs some English editing company for the betterment.

We have corrected the headings and also the whole manuscript.

Round  2

Reviewer 2 Report

Dear Authors 

Your efforts in the modification of this manuscript are highly appreciated. Please check any English spell or minor mistakes carefully. 

Check Figure captions, references and chemical names carefully.

Paper is well organized and modified according to reviewer suggestion. I am accepting this work in the present form.